# Extended POLIPHON dust conversion factor dataset for lidarderived cloud condensation nuclei and ice-nucleating particle concentration profiles

Yun He<sup>1,2,3</sup>, Goutam Choudhury<sup>4</sup>, Matthias Tesche<sup>5</sup>, Albert Ansmann<sup>6</sup>, Fan Yi<sup>1,2,3</sup>, Detlef Müller<sup>7</sup>, and Zhenping Yin<sup>7,\*</sup>

<sup>1</sup>School of Earth and Space Science and Technology, Wuhan University, Wuhan China

Abstract. Mineral dust is abundant in the atmosphere. To assess its climate impact, it is essential to obtain information on the three-dimensional distribution of cloud condensation nucleation (CCN) and ice-nucleating particle (INP) concentrations related to mineral dust. The POlarization LIdar PHOtometer Networking (POLIPHON) method uses aerosol-type-dependent conversion factors to transform lidar-derived aerosol optical parameters into CCN- and INP-relevant microphysical parameters. We present a global data set of conversion factors at 532 nm obtained using Aerosol RObotic NETwork (AERONET) observations at 137 sites for INP and 123 sites for CCN calculations. Dust presence is identified using a column-integrated dust ratio threshold of 80%, derived from AERONET columnar particle linear depolarization ratio at 1020 nm. INP-relevant conversion factors ( $c_{250,d}$ ,  $c_{s,d}$ , and  $c_{s,100,d}$ ) exhibit distinct regional patterns, generally lower near deserts and increasing downstream from dust sources. CCN-relevant conversion factors ( $c_{100,d}$  and  $\chi_d$ ) display significant site-to-site variation. A comparison of dust-related particle concentration profiles derived using both POLIPHON and the independent OMCAM (Optical Modelling of the CALIPSO Aerosol Microphysics) retrieval shows that profiles generally agree within an order of magnitude. This result is consistent with the respective retrieval uncertainties and corroborates the usefulness of lidar observations for inferring dust-related CCN and INP concentration profiles.

<sup>&</sup>lt;sup>2</sup>Key Laboratory of Geospace Environment and Geodesy, Ministry of Education, Wuhan, China.

<sup>&</sup>lt;sup>3</sup>State Observatory for Atmospheric Remote Sensing, Wuhan, China.

<sup>&</sup>lt;sup>4</sup>Department of Environment, Planning and Sustainability, Bar-Ilan University, Ramat Gan, Israel.

<sup>&</sup>lt;sup>5</sup>Leipzig Institute for Meteorology (LIM), Leipzig University, Leipzig, Germany.

<sup>&</sup>lt;sup>6</sup>Leibniz Institute for Tropospheric Research (TROPOS), Leipzig, Germany

<sup>&</sup>lt;sup>7</sup>School of Remote Sensing and Information Engineering, Wuhan University, Wuhan, China.

<sup>\*</sup>Correspondence to: Zhenping Yin (zp.yin@whu.edu.cn)

## 1. Introduction

Aerosol-cloud interactions (ACI) contribute the largest uncertainty in our current understanding of global climate change (IPCC, 2021). To study ACIs, it is essential to link characteristic parameters of both aerosols and clouds. Parameters such as cloud phase, cloud fraction, ice/liquid water content, and the size and number concentrations of ice crystals and liquid droplets are typically used for estimating the climate effect of clouds (Huang et al., 2006; Rosenfeld et al., 2014). Estimates of the climate effect of aerosols are often based on aerosol optical depth (AOD), aerosol index, or aerosol number concentration (Nakajima et al., 2001; Rosenfeld, 2006; Zhao et al., 2019). A better assessment of ACI effects requires information on the number concentration of cloud-relevant aerosol particles at cloud level, particularly of ice nucleating particles (INP) and cloud condensation nuclei (CCN) (Kanji et al., 2017; Korolev et al., 2017).

The POLIPHON (POlarization LIdar PHOtometer Networking) method has been developed for inferring INP and CCN number concentration profiles from ground-based lidar measurements (Mamouri and Ansmann, 2014, 2015; Mamouri et al., 2016). It has also been applied to lidar observations from space (Marinou et al., 2019; Choudhury et al., 2022). This method combines polarization lidar observations with sun photometer measurements, meteorological parameters (from reanalysis or radiosonde data), and aerosol-type specific parameterizations to retrieve profiles of INP concentrations (INPC) and CCN concentrations (CCNC). Therefore, the POLIPHON method holds potential for global application, ranging from individual or multiple ground-based lidar sites (Ansmann et al., 2019a, 2019b; Haarig et al., 2019; Marinou et al., 2019; Hofer et al., 2020; He et al., 2021b) to spaceborne lidar observations, such as CALIOP (Cloud-Aerosol Lidar with Orthogonal Polarization) (Winker et al., 2009; Georgoulias et al., 2020; He et al., 2022; Shen et al., 2024) and ongoing EarthCARE mission (Wehr et al., 2023), and ground-based lidar networks (Baars et al., 2016; Pappalardo et al., 2014).

An essential step of POLIPHON is the transformation of lidar-derived aerosol-type-specific extinction coefficients to particle number concentrations (with particle size above a certain threshold) and particle surface area concentrations as input to INP- and CCN-parameterizations with the help of related conversion factors (Ansmann et al., 2019a; He et al., 2021b, 2023). However, for each aerosol type, conversion factors can vary from region to region due to differences in particle microphysics. The global application of POLIPHON therefore requires spatially resolved information about these conversion factors.

Dust aerosols are of particular importance as they mark a major contributor to global INP and CCN burden (Kanji et al., 2017; Choudhury and Tesche, 2022a; Casquero-Vera et al., 2023; Chatziparaschos et al., 2024; Herbert et al., 2025). The most challenging aspect of deriving dust-related conversion factors is identifying the presence of dust in sun photometer observations, such as in the framework of the Aerosol Robotic Network (AERONET, Holben et al., 1998; Giles et al., 2023). So far, POLIPHON studies have used an Ångström exponent (AE, for 440–870 nm) <0.3 and AOD at 532 nm >0.1 (Ansmann, et al., 2019a) or a column-integrated dust ratio >53% (based on the 1020-nm particle linear depolarization ratio) (He et al., 2023) for identifying dust-dominated observations. Here we aim to extend the earlier work on dust-related conversion factors to additional AERONET sites that cover most regions on Earth where local or transported dust aerosols are likely to occur.

The extended conversion-factor dataset can be applied to retrieving dust-related CCNC and INPC profiles that can be compared to independent datasets or measurements. The uncertainties in POLIPHON-derived INPC are primarily caused by the considered INP parameterizations (DeMott et al., 2015; Ullrich et al., 2017). Those are highly dependent on meteorological parameters, which makes INPC comparison a very challenging task. In contrast, CCN parameterizations are much simpler (Shinozuka et al., 2015) and easily applicable in a validation study. Therefore, we will compare dust-related CCNC profiles derived from spaceborne CALIOP observations using POLIPHON with those obtained by the Optical Modelling of the CALIPSO Aerosol Microphysics (OMCAM, Choudhury and Tesche, 2022a, 2022b, 2023a) retrieval. OMCAM assumes that each aerosol type can be represented by a single particle size distribution (PSD). This fundamental difference to POLIPHON provides us with a unique opportunity to examine the potential influence (sensitivity to retrieving uncertainty) of such an assumption in CCNC retrievals.

The paper is organized as follows. We first introduce the POLIPHON method, the process for retrieving dust-related

conversion factors, and the OMCAM algorithm. Section 3 presents the derived dataset of dust-related conversion factors. In Section 4, we conduct a dust-related CCN profile comparison study between the POLIPHON and OMCAM methods. The main findings of the study are summarized in Section 5.

## 2. Data and methodology

#### 2.1 POLIPHON method for dust-related CCN and INP retrieval

POLIPHON was developed for deriving height-resolved aerosol-type-specific information on particle mass, INPC, and CCNC based on measurements with polarization lidar and sun photometer (Mamouri and Ansmann, 2014, 2015; Mamouri et al., 2016). The method is considered particularly reliable in the presence of mineral dust (Hofer et al., 2020; Ansmann et al., 2021b; He et al., 2021b) due to the large particle linear depolarization ratio of non-spherical dust particles (Tesche et al., 2009).

Table 1. Overview of the computation of dust-related mass, INP, and CCN concentrations using the POLIPHON method based on polarization lidar observations (Tesche et al., 2010; Ansmann et al., 2019a). The subscripts 'p', 'd', and 'nd' denote 'particle', 'dust', and 'non-dust', respectively.

| Main task                                                                                                         | Input parameter                       | Calculation                                                                                                                                                                     | Uncertainties |
|-------------------------------------------------------------------------------------------------------------------|---------------------------------------|---------------------------------------------------------------------------------------------------------------------------------------------------------------------------------|---------------|
| Divide lidar-derived particle backscatter $\beta$ into dust $\beta_d$                                             | $\beta(z), \delta_{\mathrm{p}}(z)$    | $\beta_{\rm d}(z) = \beta(z) \frac{\left(\delta_{\rm p}(z) - \delta_{\rm nd}\right)(1 + \delta_{\rm d})}{\left(\delta_{\rm d} - \delta_{\rm nd}\right)(1 + \delta_{\rm p}(z))}$ | 10-20%        |
| and non-dust $eta_{ m nd}$                                                                                        | $\beta_{\rm d}(z)$ , dust lidar ratio | $\alpha_{\rm d}(z) = LR \cdot \beta_{\rm d}(z)$                                                                                                                                 | 15-25%        |
| Convert into dust mass                                                                                            | $\alpha_{\rm d}(z)$ , dust-           | $M_{\rm d}(z) = c_{\rm v,d} \times \rho_{\rm d} \times \alpha_{\rm d}(z)$                                                                                                       | 20-30%        |
| concentration, CCN- and INP-relevant parameters: dust number concentration $n_{250,d}$ and $n_{100,d}$ , and dust | related<br>conversion<br>factors      | $n_{250,d}(z) = c_{250,d} \times \alpha_{d}(z)$                                                                                                                                 | 25-35%        |
|                                                                                                                   |                                       | $s_{\rm d}(z) = c_{\rm s,d} \times \alpha_{\rm d}(z)$                                                                                                                           | 30-40%        |
|                                                                                                                   |                                       | $s_{100,d}(z) = c_{s,100,d} \times \alpha_{d}(z)$                                                                                                                               | 20-30%        |
| particle surface area concentration $s_d$ and $s_{100,d}$                                                         |                                       | $\log(n_{100,d}(z)) = \log(c_{100,d}) + \chi_d \log(\alpha_d(z))$                                                                                                               | 50-200%       |
| Input parameters $n_{250,d}$ , $s_d$ , $s_{100,d}$ , and $n_{100,d}$ into different INP and CCN parameterizations | $n_{250,d}(z), T(z)$                  | INP parameterization D-15 (DeMott et al., 2015)                                                                                                                                 | 50-300%       |
|                                                                                                                   | $s_{\rm d}(z), T(z)$                  | INP parameterization U-17d (Ulrich et al., 2017)                                                                                                                                | 50-300%       |
|                                                                                                                   | $s_{100,d}(z), T(z)$                  | INP parameterization U-17d (Ulrich et al., 2017)                                                                                                                                | 50-300%       |
| •                                                                                                                 | $n_{100, d}(z)$                       | CCN parameterization: $n_{\text{CCN,d}}(z) = f_{\text{ss,d}} \times n_{100,d}(z)$ (Shinozuka et al., 2015)                                                                      | 50-200%       |

The processing steps of POLIPHON are summarized in Table 1. The method starts with the retrieval of the particle backscatter coefficient  $\beta_p$  from lidar observations using the method of Fernald (1984). This parameter is separated into contributions from dust and non-dust, i.e.,  $\beta_d$  and  $\beta_{nd}$  (Tesche et al., 2009). Next, the dust extinction coefficient  $\alpha_d$  is obtained by multiplying  $\beta_d$  with a dust lidar ratio (LR) of 30-60 sr (Müller et al., 2007; Tesche et al., 2011; Hofer et al., 2017; Hu et al., 2020; Peng et al., 2021; Floutsi et al., 2023). Dust particles originating from different deserts can exhibit distinct optical and microphysical properties, such as particle size distribution and complex refractive index; additionally, variations in dust transport pathways may lead to differences in aging, mixing, and removal processes. These factors may result in regional-variation of the dust lidar ratio. The derived  $\alpha_d$  is then converted into

- the concentration of particles with radii larger than 100 nm  $(n_{100.d})$  for the CCN retrieval,
- the concentration of particles with radii larger than 250 nm  $(n_{250,d})$  for the INP retrieval, and
- the surface area concentration  $s_d$  and  $s_{100,d}$  for the INP retrieval

with the help of the corresponding conversion factors, i.e.,  $c_{100,d}$ ,  $\chi_d$ ,  $c_{250,d}$ ,  $c_{s,d}$ , and  $c_{s,100,d}$  (Mamouri et al., 2016; Ansmann et al., 2019a). It should be noted that, for retrieving the CCN-relevant parameter  $n_{100,d}$ , a log-log regression analysis is applied, in which the conversion factor  $c_{100,d}$  and regression coefficient  $\chi_d$  are determined (Shinozuka et al., 2015). Finally,  $n_{250,d}$ ,  $s_d$ ,

and  $s_{100,d}$  are used as input for various dust INP parameterization schemes (DeMott et al., 2015; Ulrich et al., 2017) to derive the dust-related INP profile  $n_{\text{INP,d}}(z)$ .  $n_{100,d}$  is used to obtain the dust-related CCN profile  $n_{\text{CCN,d}}(z)$  following Shinozuka et al. (2015) as:

$$n_{\text{CCN,d}}(z) = f_{\text{ss,d}} \times n_{100,d}(z) \tag{1}$$

where  $f_{\rm ss,d}$  is the water supersaturation-dependent factor, with values of 1.00, 1.35, and 1.70 for supersaturations of 0.15-0.20%, 0.25%, and 0.40%, respectively. Note that for retrieving CCN and INP,  $n_{\rm 250,d}$ ,  $s_{\rm d}$ , and  $n_{\rm 100,d}$  under dry conditions are needed. Here dust is considered hydrophobic so an additional correction is not necessary (Mamouri et al., 2016).

In addition, from the dust extinction coefficient, we can also derive the dust mass concentration profile  $M_d(z)$  by using the extinction-to-volume conversion factor  $c_{v,d}$  and an assumed dust density  $\rho_d$  with the following equation (Jing et al., 2024):

$$M_{\rm d}(z) = \rho_{\rm d} \times \alpha_{\rm d}(z) \times c_{\rm v,d} \tag{2}$$

We assume  $\rho_d$  to be 2.6 g cm<sup>-3</sup> (Ansmann et al., 2019a). The parameters  $c_{v,d}$  and  $\rho_d$  together determine the so-called mass extinction efficiency (Wang et al., 2021). Detailed computational procedures, associated equations, and uncertainty analyses are provided in Mamouri and Ansmann (2015) and Ansmann et al. (2019a).

# 2.2 Conversion factors derived from AERONET dataset

The conversion factors in the POLIPHON method are dependent on both aerosol type and geographic region (Ansmann et al., 2019a). In this section, we describe the retrieval of  $c_{v,d}$ ,  $c_{100,d}$ ,  $\chi_d$ ,  $c_{250,d}$ , and  $c_{s,d}$ . To ensure consistency with Ansmann et al. (2019a), we also present the conversion factor  $c_{s,100,d}$  for calculating the surface area concentration of dust particles with radii larger than 100 nm. These conversion factors are derived from AERONET measurements of AOD at eight wavelengths (i.e., 340, 380, 440, 500, 675, 870, 1020, and 1064 nm) (Holben et al., 1998; Giles et al., 2019) and the particle size distributions provided in the aerosol inversion data product (Sinyuk et al., 2020) as illustrated in Figure 1. The first step is identifying the presence of dust in an observation. We use the columnar particle linear depolarization ratio (PLDR) at 1020 nm  $\delta_{1020nm}^p$  from the AERONET inversion product for identifying dust data points (Noh et al., 2017; Shin et al., 2018, 2019; He et al., 2023). Due to the spheroid particle assumption in the AERONET algorithm, PLDR at the near-infrared wavelength may show some overestimations as compared with polarization lidar observations (Toledano et al., 2019; Haarig et al., 2022). Nevertheless, its polarization sensitivity is sufficient for identifying nonspherical particles. Dust is the primary nonspherical particle in the atmosphere; thus, we consider other potential types of nonspherical aerosols, such as fresh smoke, volcanic ash, and pollen as secondary.

We calculate the column-integrated dust ratio  $R_{\rm d,1020nm}$  following Shin et al. (2019):

$$R_{\rm d,1020nm} = \frac{\left(\delta_{\rm 1020nm}^{\rm p} - \delta_{\rm nd}^{\rm p}\right)\left(1 + \delta_{\rm d}^{\rm p}\right)}{\left(\delta_{\rm d}^{\rm p} - \delta_{\rm nd}^{\rm p}\right)\left(1 + \delta_{\rm 1020nm}^{\rm p}\right)} \tag{3}$$

where the dust  $\delta_{\rm d}^{\rm p}$  and non-dust  $\delta_{\rm nd}^{\rm p}$  PLDR values are set to 0.30 and 0.02, respectively. Within the atmospheric column,  $R_{\rm d,1020nm}$  reflects the contribution of dust to the total particle backscatter coefficient of an external aerosol mixture (Tesche et al., 2009). For reference, lidar observations of the PLDR of pure dust range between 0.30 and 0.35 (Freudenthaler et al., 2009; Floutsi et al., 2023). In this extended study, we use  $R_{\rm d,1020nm} \geq 80\%$  as a criterion for identifying the 'dust-presence' data points, which are subsequently used to calculate the conversion factors for dust aerosols. We have conducted a sensitivity analysis by adjusting this threshold value for the column-integrated dust ratio that we used to identify dust-containing data points. Varying this criterion largely affects the number of AERONET sites for which conversion factors are available. The selection of an optimal threshold involves balancing data availability and closeness to pure dust conditions. The use of  $R_{\rm d,1020nm} \geq 80\%$  marks a compromise between the identification of pure dust cases with  $0.89 

Figure 1. Flow chart of calculating the dust-related POLIPHON conversion factors based on the AERONET measurements, including the Version 3 Level-1.5 or -2.0 aerosol inversion product (corresponding to the finally obtained Level-1.5 or -2.0 conversion factor dataset, respectively) (Sinyuk et al., 2020; AERONET, 2023b) and Level-2.0 AOD product (Giles et al., 2019; AERONET, 2022a). The selection scheme of dust-containing data points refers to Shin et al. (2018, 2019).

Figure 2 shows the AERONET sites selected for retrieving the dust-related conversion factors in this study. We only include AERONET sites with valid data spanning observations of more than two years before October 2022 (AERONET, 2023a, 2023b). In total, 198 AERONET sites are included, geographically covering most desert regions and the major transport pathways of dust plumes (Hu et al., 2019; Mona et al., 2023). The origin of dust particles at each site can be quite variable. In the tropic and mid-latitude of the Northern Hemisphere, a majority of dust particles generally appears along the dust belt that spans the Saharan Desert, Middle East deserts, Asian deserts (mainly the Taklimakan Desert and Gobi Desert), and their downstream regions (Hofer et al., 2017). The high-latitude dust of the Northern Hemisphere can be contributed by high-latitude local Aeolian dust origins (Bullard et al., 2016) as well as south-to-north meridional transport originating from Asian and African deserts (Shi et al., 2022). In the Southern Hemisphere, there are major dust sources including the Patagonian Desert in South America, Australia's deserts, and the Kalahari Desert in Southern Africa. In addition, anthropogenic dust from agriculture, transportation, or construction, can also play a significant role (Chen et al., 2023).

Figure 2. Overview of AERONET sites used for inferring dust-related POLIPHON conversion factors. The orange crosses show the locations of near-desert, oceanic, and coastal sites in He et al. (2023). The solid circles in different colors indicate the locations of 198 AERONET sites in North America (dark red), South America (red), Africa (modena), Europe (blue), North and East Asia (lilac), South and West Asia (magenta), and Australia (green).

#### 2.3 OMCAM algorithm for retrieving CCN concentration

165

170

175

180

185

190

Choudhury and Tesche (2022a) developed the OMCAM algorithm to derive global, height-resolved aerosol-type-specific CCNC from spaceborne CALIPSO (Winker et al., 2009) lidar observations. To calculate dust-related CCNC, they obtain dust-related backscatter and extinction coefficients from three aerosol mixtures in the CALIOP level-2 aerosol profile product, namely mineral dust, polluted dust, and dusty marine, following Tesche et al. (2009). The CALIPSO aerosol model provides microphysical properties of each aerosol type included in the retrieval (Omar et al., 2009). It provides a dust-specific normalized volume size distribution ( $V_{\rm d,normalized}$ ) and refractive index which is used to obtain the corresponding dust extinction coefficient at 532 nm ( $\alpha_{\rm d,normalized}$ ), through light-scattering calculations (Gasteiger and Wiegner, 2018). The ratio  $V_{\rm t}$  of the CALIOP-measured dust extinction  $\alpha_{\rm d,measured}$  and  $\alpha_{\rm d,normalized}$  is used to scale the normalized volume size distribution to obtain:

$$V_{\rm d,scaled} = V_{\rm t} \times V_{\rm d,normalized} \tag{5}$$

This scaled size distribution  $V_{d,scaled}$  is the one that best reproduces the dust extinction coefficient provided in the CALIPSO aerosol profile product. Converting  $V_{d,scaled}$  into a number size distribution and using Eq. (1) leads to the dust-related CCNC profiles.

The instantaneous and gridded OMCAM-derived CCNC are found to be consistent with independent in-situ measurements (Choudhury and Tesche, 2022b; Choudhury et al., 2022; Aravindhavel et al., 2023) and reanalysis results (Choudhury et al., 2025). They are also used for studying aerosol-cloud interactions for warm and cold clouds based on spaceborne observations (Alexandri et al., 2024). Choudhury and Tesche (2023a) applied the OMCAM algorithm to generate the first global 3-D CCNC dataset using more than 15 years of CALIOP Level-2 aerosol profile products. This CCNC dataset includes five aerosol subtypes, i.e., marine, dust, polluted continental, clean continental, and elevated smoke. It is available at a uniform latitude-longitude grid of 2°×5° with a temporal resolution of one month.

## 2.4 Scheme of comparing dust CCNC from POLIPHON and OMCAM

There are several algorithms for retrieving CCNC profiles from lidar observations that all hinge on the assumed parameters of the PSD. Those methods are generally based on multiwavelength lidar data and might consult look-up tables (Lv et al., 2018; Zhou et al., 2024), in-situ measurements (Tan et al., 2019), or machine learning (Redeman and Gao, 2024) to convert optical data to microphysical parameters and offer the advantage of considering realistic and variable PSD estimates (Müller et al., 2014). However, the instrumental complexity required to obtain the data used in the abovementioned methods has so far

ruled out spaceborne application.

195

200

205

OMCAM has been designed for retrieving aerosol-type-specific CCNC from spaceborne lidar observations. POLIPHON has initially been developed for retrieving aerosol-type-specific CCNC and INPC based on ground-based lidar observations, and has subsequently been extended to spaceborne applications (Marinou et al., 2019). Therefore, a key difference between the OMCAM and POLIPHON methods is that OMCAM only employs a fixed shape of aerosol-type-specific PSD from CALIPSO's aerosol model, whereas POLIPHON considers the use of regionally varying aerosol-type-specific PSDs (i.e., conversion factors that are calculated from PSDs). In particular, regional variations in dust PSD mainly result from the deposition of dust particles during their long-range transport (Ansmann et al., 2017; Rittmeister et al., 2017). Coarse dust particles generally deposit prior to fine dust particles within the plume along the transport pathway from their dust sources (Ratcliffe et al., 2024), which causes variations in dust PSD. By comparing dust CCNC results obtained from these two methods, this study also provides an opportunity to evaluate whether employing a fixed dust PSD is sufficient for deriving the global dust CCNC distribution or whether regionally dependent dust PSDs are necessary (Adebiyi et al., 2023).

In this study, the monthly dust-related CCNC profiles obtained via POLIPHON and from the OMCAM climatology (Choudhury and Tesche, 2023b) are compared for selected AERONET sites. For consistency, both methods consider monthly dust-specific extinction coefficient profiles from the CALIOP Level 2 profile product for inferring CCNC in grid boxes closest to the considered AERONET stations. First, the CALIOP Version 4.20 Level-2 aerosol profile product (Omar et al., 2009) undergoes several data quality control procedures, as listed in Section 3.1.1 of Choudhury and Tesche (2023a). Next, we separate dust backscatter coefficient profiles from the aerosol subtypes of dust, polluted dust, and dusty marine using the method of Tesche et al. (2009). These dust backscatter coefficient profiles, combined with an assumed dust lidar ratio of 44 sr (Kim et al., 2018), are then used to form a global gridded (latitude: 2°, longitude: 5°) monthly-average dust extinction coefficient dataset, with a vertical resolution of 60 m from the surface to an altitude of 8 km. This 3-D global dust extinction dataset, derived from the CALIOP data spanning June 2006 to December 2021 (except for February 2016 due to the unavailability of CALIOP data), serves as inputs for retrieving dust-related CCNC implementing with both the POLIPHON and OMCAM methods.

## 3. Global distribution of dust-related conversion factors

Figure 3 presents dust-related conversion factors at four (out of 198) typical city sites, i.e., Beijing (China), KAUST\_Campus (Saudi Arabia), La\_Parguera (Puerto Rico), and Granada (Spain). Note that we use the formal site names defined by AERONET. The large difference in the number of dust-presence data points is due to both the duration of sun photometer observations and the frequency and extent of dust intrusions. The local atmospheric environment varies from site to site, as indicated by the averaged 532-nm AODs of 0.438, 0.374, 0.130, and 0.132 for Beijing, KAUST\_Campus, La\_Parguera, and Granada, respectively. Beijing shows larger  $c_{v,d}$  and smaller  $c_{250,d}$  compared to the other sites, which is probably due to the influence of more local pollutants (usually smaller size with larger concentration). This is discussed further below.

Figure 3. Relationship between the 532-nm aerosol extinction coefficient and particle (radius >250 nm) number concentration  $n_{250,d}$ , volume concentration  $v_d$ , and surface area concentration  $s_d$  and  $s_{100,d}$  (radius >100 nm) for dust-presence data points (number denoted by N) at four typical city sites, i.e., (a) and (e) for Beijing (39.98°N, 116.38°E), (b) and (f) for KAUST\_Campus (22.30°N, 39.10°E), (c) and (g) for La\_Parguera (17.97°N, 67.05°W), and (d) and (h) for Granada (37.16°N, 3.61°W). The corresponding dust-related conversion factors are provided in the corresponding panels.

Figure 4 presents the global distribution of dust-related mass concentration and INP-relevant conversion factors  $c_{v,d}$ ,  $c_{250,d}$ ,  $c_{s,d}$  and  $c_{s,100,d}$ . Ansmann et al. (2019) applied a threshold value of AOD>0.1 (i.e., aerosol extinction >100 Mm<sup>-1</sup>) when calculating INP- and CCN-related dust conversion factors, and thereby excluded cases with very clean atmospheric conditions that may introduce large uncertainties in the computations. He et al. (2023) attempted to relax this lower limit to an AOD of 0.02 (i.e., aerosol extinction >20 Mm<sup>-1</sup>) to increase the number of available data points and eliminate abnormal values. Based on the comparisons with Ansmann et al. (2019), we find that loosening relaxing this threshold value does not significantly affect the results. Therefore, we only considered data points with aerosol extinctions exceeding 20 Mm<sup>-1</sup>. For a site to be considered, it had to show at least 15 valid dust-presence data points, which applies to 137 out of the 198 selected AERONET sites. The regional variation of the conversion factors reflects the distinct microphysical properties of dust along its transport pathways and the varying impact of mixing with other aerosol types (Philip et al., 2017). Moreover, dust from different deserts may exhibit different microphysical properties. Particularly changes in the dust PSD contribute to the regional variation of conversion factors.

As shown in Figure 4a, the extinction-to-volume conversion factor  $c_{v,d}$  ranges from  $0.4 \times 10^{-12}$  to  $0.8 \times 10^{-12}$  Mm·m³·m³ over the dust belt region of the Northern Hemisphere (e.g., North Africa, the Middle East, and Central Asia) as well as the major downstream regions of dust transport (Europe, East Asia, and Western America). In addition, similar  $c_{v,d}$  values are also obtained in some Australian sites impacted by local desert dust. Values of  $c_{v,d}$  generally decrease along the routes of dust transport due to the removal of dust particles by gravitation settling and cloud processing as well as the mixing with other, usually smaller and more spherical aerosols. This is consistent with the dust-related conversion factors found at Lanzhou near the deserts in East Asia and Wuhan far away from deserts (He et al., 2021b). Kai et al. (2023) observed a decreasing trend in the dust mass-extinction conversion factor along the transport pathway of dust aerosols originating from the Gobi Desert, which suggests an equivalent decreasing trend in  $c_{v,d}$ , if assuming a fixed dust density. The larger geographical coverage of

the data set presented here provides valuable information for global dust models in which mass extinction efficiency is a key parameter (Adebiyi et al., 2020; Han et al., 2022). In contrast, Figure 4b shows that  $c_{250,d}$  near desert regions are relatively lower compared to polluted regions downstream of deserts. Notably, a gradual increase in  $c_{250,d}$  is evident when following the meridional transport of dust from North Africa to Northern Europe, corresponding to the typical northward transport pathway of Saharan dust. Generally,  $c_{s,d}$  and  $c_{s,100,d}$  show slightly higher values at the sites far from desert regions. Moreover, these two factors are more sensitive to the presence of other aerosols (He et al., 2023), which may explain the larger site-to-site variation.

Figure 4. POLIPHON dust-related conversion factors  $c_{v,d}$ ,  $c_{250,d}$ ,  $c_{s,d}$  and  $c_{s,100,d}$  obtained from dust data points ( $R_{d,1020nm} \ge 80\%$ ) at 137 AERONET sites.

Figure 5 shows the derived CCN-relevant conversion factors  $c_{100,d}$  and  $\chi_d$  at four typical city sites, i.e., the same as those in Figure 3. Note that only data points with aerosol extinctions between 20 Mm<sup>-1</sup> and 600 Mm<sup>-1</sup> are considered in the calculations. The correlations at Beijing, Granada, and La\_Parguera are generally strong; however, at KAUST\_Campus, the data points tend to be scattered as the aerosol extinction coefficient increases, indicating a growing influence of local non-dust particles, e.g. anthropogenic pollutants.

Figure 5. Relationship between aerosol extinction coefficient at 532 nm and aerosol particle number concentration  $n_{100,d}$  (radius >100 nm) for dust-presence data points at the same sites as Figure 3. The corresponding dust-related conversion factors  $c_{100,d}$  and  $\chi_d$  are provided.

Figure 6 presents the distribution of worldwide  $c_{100,d}$  and  $\chi_d$  values. When applying the regression analysis, Ansmann et al. (2019a) found that the relationship becomes much weaker when the 532-nm AOD exceeds 0.6 (i.e., extinction coefficient >600 Mm<sup>-1</sup>); the authors provide a thorough discussion on this issue (see Section 3.2 therein). Accordingly, we allocated extinction coefficients between 20 Mm<sup>-1</sup> to 600 Mm<sup>-1</sup> to the identified dust-containing data points and calculated the CCN-related conversion factors. For each site, the conversion factors are considered only if at least 15 valid dust-presence data points are available. As a result, 123 out of the 198 selected AERONET sites have valid conversion factors. No identifiable regional variation pattern is observed for  $c_{100,d}$  and  $\chi_d$ , indicating that these factors are more sensitive to the contribution of local fine-mode particles. This suggests that to derive dust-related CCNC, it is crucial to use region-specific conversion factors rather than relying on a global average, which is consistent with the results given by Ansmann et al. (2019a).

Figure 6. POLIPHON dust-related conversion factor  $c_{100,d}$  and associated regression coefficient  $\chi_d$  obtained from dust data points  $(R_{d,1020nm} \ge 80\%)$  at 123 AERONET sites.

The conversion factors presented in Figures 4 and 6 can be accessed at <a href="https://doi.org/10.5281/zenodo.16781089">https://doi.org/10.5281/zenodo.16781089</a> (He, 2025). Note that a conversion factor is provided only when the corresponding number of identified dust-presence data points exceeds 15. Considering the use of dust-dominant mixture (with a columnar dust ratio  $R_{d,1020nm} \ge 80\%$ ) in the calculation, traces of local non-dust components (e.g., anthropogenic pollutants) may be included. Therefore, a larger dust-presence data point number and a smaller standard deviation indicate that the corresponding conversion factors more closely represent the local dust properties. Note that the results from the dust belt region of the Northern Hemisphere are considered more reliable (Hofer et al., 2017), as they typically involve over 1000 dust-presence data points. In contrast, the results from downwind sites located in more remote regions of dust transport, such as North and South America, likely reflect occasional dust intrusion events (long-range transport), meaning that the derived conversion factors may not be representative from a statistical point of view, and thus, we recommend further validations by in-situ measurements. We have also endeavored to compile a gridded dust conversion-factor dataset for expedient future use in studying global ACI using gridded spaceborne lidar datasets. However, this has proven challenging due to the limited number of available sites in comparison to global coverage and their inhomogeneous geographical distribution. Therefore, when applying this conversion factor dataset, we recommend selecting values from the nearest available site.

## 4. Comparing dust-related CCN concentrations from POLIPHON and OMCAM

We verify the extended conversion-factor dataset by comparing the obtained dust-related CCNC profiles with OMCAMderived CCN data (Choudhury and Tesche, 2022a, 2023a) for 12 AERONET sites. The sites were selected to provide a wide geographical spread and to cover the range of  $\chi_d$  from 0.7 to 1.1. Table 2 gives an overview of those sites and the inferred parameters. More details can be found in the dataset (He, 2025). The CCNC values from geographical grids containing the selected AERONET sites are extracted for comparison. It should be noted that dust particles are typically hydrophobic; however, they may undergo aging processes during their transport, which may change their surface properties and make them capable of acting as CCNs.

Table 2. Overview of the AERONET sites used for comparing the dust-related CCNC from POLIPHON and OMCAM. The total number of data points for each site is derived from the AERONET Level-1.5 aerosol inversion product.

|               | AERONET site name | City name                | Location          | $c_{100,d}$ (cm <sup>-3</sup> for $\alpha_d$ =1 Mm <sup>-1</sup> ) | $\chi_{ m d}$ | Dust-presence / total data point number |
|---------------|-------------------|--------------------------|-------------------|--------------------------------------------------------------------|---------------|-----------------------------------------|
| North America | La_Parguera       | La Parguera, Puerto Rico | 17.97°N, 67.05°W  | 1.68±1.02                                                          | 0.81±0.01     | 2005/12068                              |
|               | Fresno_2          | Fresno, USA              | 36.79°N, 119.77°W | 3.21±1.19                                                          | 0.84±0.08     | 149/105535                              |
| South America | CEILAP-BA         | Buenos Aires, Argentina  | 34.56°S, 58.51°W  | 3.25±1.18                                                          | 0.93±0.07     | 123/11433                               |
|               | Trelew            | Trelew, Argentina        | 43.25°S, 65.31°W  | 1.41±1.22                                                          | 0.81±0.07     | 133/9240                                |
| Africa        | Cairo_EMA_2       | Cairo, Egypt             | 30.08°N, 31.29°E  | 2.79±1.03                                                          | 0.76±0.03     | 1959/11919                              |
|               | IER_Cinzana       | IER-Cinzana, Mali        | 13.28°N, 5.93°W   | 2.11±1.06                                                          | 0.82±0.04     | 10633/14269                             |
|               | Zinder_Airport    | Zinder, Niger            | 13.77°N, 8.99°E   | 3.01±1.04                                                          | 0.92±0.03     | 7063/9867                               |
| Middle East   | KAUST_Campus      | Thuwal, Saudi Arabia     | 22.30°N, 39.10°E  | 3.27±1.02                                                          | 0.90±0.02     | 5294/17022                              |
| East Asia     | Beijing           | Beijing, China           | 39.98°N, 116.38°E | 3.52±1.15                                                          | 1.01±0.10     | 545/14409                               |
| Australia     | Birdsville        | Birdsville, Australia    | 25.90°S, 139.35°E | 2.34±1.14                                                          | 1.05±0.05     | 371/11569                               |
| Europe        | Granada           | Granada, Spain           | 37.16°N, 3.61°W   | 2.76±1.03                                                          | 0.70±0.02     | 2670/20983                              |
|               | Palma_de_Mallorca | Palma de Mallorca, Spain | 39.55°N, 2.63°E   | 3.59±1.09                                                          | 0.85±0.03     | 1207/11652                              |

Figure 7 presents the average dust-related CCNC profiles at a supersaturation of 0.2% with respect to liquid water from the POLIPHON and OMCAM methods, respectively, at the 12 sites listed in Table 2. CCNC values from OMCAM are generally larger than those from POLIPHON with a difference of less than one order of magnitude. The overall uncertainty in OMCAM-derived CCNC is estimated to be 200%-300% (Choudhury and Tesche, 2023a), whereas the uncertainty in POLIPHON-derived CCNC ranges from 50% to 200% (Ansmann et al., 2019a). As a result, even differences as large as an order of magnitude can still be considered acceptable within the uncertainty bound, particularly for a parameter like CCNC, which can vary by more than five orders of magnitude at a given location (Choudhury and Tesche, 2022b).

Figure 7. Dust-related CCNC (at a water supersaturation ss=0.2%) profiles derived using POLIPHON (red) and OMCAM (blue) at 12 selected AERONET sites. Profiles represent the average from June 2006 to December 2021 and are based on monthly means.

The comparison also suggests that the globally fixed dust PSD defined by the CALIPSO aerosol model may not accurately depict the dust microphysical properties at these locations. The site-average dust-related PSDs in Figure 8a highlight the dominance of coarse-mode particles at the selected sites though significant differences in maximum concentration are visible between sites. Figure 8b presents the normalized particle volume size distribution provided in the CALIPSO aerosol model, which is used in OMCAM retrieval, together with the twelve-site average particle volume size distributions for different dust identification schemes ( $R_{d,1020nm} \ge 80\%$  and 89%). If the dust fraction in the atmospheric column increases, the number of coarse-mode dust particles rapidly increase. Differences between CCNC values from POLIPHON and OMCAM may arise from site-to-site variations of dust microphysical properties, such as particle size distribution, refractive index, and lidar ratio due to gravitational deposition along the dust transport pathway from dust sources (Ansmann et al., 2017; Ratcliffe et al., 2024). Compared with the average PSDs of identified dust data (see Figure 8b), the CALIPSO aerosol model dust PSD exhibits a similar mean radius for both the fine and coarse modes ( $\mu_f$  and  $\mu_c$ ). However, significant differences are observed in the volume fraction of the coarse and fine mode ( $v_f$  and  $v_c$ ) between the identified dust PSDs and the fixed normalized dust PSD from the CALIPSO aerosol model. It is evident that the coarse-to-fine mode particle ratios are much higher for the identified dust data in this study compared to those from the CALIPSO aerosol model applied in OMCAM. This implies that when using the fixed dust PSD in OMCAM to reproduce CALIOP-derived aerosol extinction, a larger number concentration of fine-mode dust particles is produced, leading to a higher  $n_{100,d}$ , and thus, a higher dust CCNC value. Moreover, the varying influence of local aerosols is also contributed since dust-dominant mixture data points from AERONET are applied in this study.

Figure 8. (a) Dust column-integrated particle volume size distributions at the selected AERONET sites from AERONET aerosol inversion data product identified using the columnar dust ratio  $R_{\rm d,1020nm}$  threshold of  $\geq$  80%. (b) Average column-integrated particle volume size distributions of the twelve selected sites with the columnar dust ratio  $R_{\rm d,1020nm}$  thresholds of  $\geq$  80% (in magenta) and  $\geq$  89%. (in cyan, as used in He et al. (2023)), and the normalized particle volume size distribution for dust from the CALIPSO aerosol model (in blue), which is used to reproduce the CALIOP-derived dust extinctions by multiplying a scaling factor in OMCAM. The standard deviations ( $\sigma_{\rm f}$  and  $\sigma_{\rm c}$ ) are 1.4813  $\mu$ m and 1.9078  $\mu$ m for fine and coarse mode, respectively; the volume fractions ( $\nu_{\rm f}$  and  $\nu_{\rm c}$ ) are 0.223 and 0.777 for fine and coarse mode, respectively; the mean radii ( $\mu_{\rm f}$  and  $\mu_{\rm c}$ ) are 0.1165  $\mu$ m and 2.8329  $\mu$ m for fine and coarse mode, respectively (Choudhury and Tesche, 2023a).

The current Version 4 CALIOP retrievals rely on globally constant, aerosol-type-specific lidar ratios that are directly linked to fixed, associated normalized PSDs (Kim et al., 2018). Therefore, incorporating the aforementioned variations into a CCNC-retrieval algorithm for CALIOP is challenging, since these normalized PSDs can only be scaled, without modifying their shape or the coarse-to-fine particle number ratios. This emphasizes the need for in-situ and remote sensing campaigns measuring dust aerosols across different regions (Ansmann et al., 2009; Ryder et al., 2013, 2018; Weinzierl et al., 2009, 2017; Haarig et al., 2017). Recent measurements from the past 15 years have not been incorporated into the CALIPSO aerosol model (Omar et al., 2009), underscoring the regional complexity of dust aerosols and suggesting that coarse-mode dust particles may be underestimated in the current model (Ansmann et al., 2017; Kok et al., 2021; Adebiyi et al., 2023; Ratcliffe et al., 2024). This conclusion is consistent with the results shown in Figure 8, suggesting an underestimation of the coarse-to-fine dust particle number ratio. The upcoming Version 5 CALIOP data product is expected to include regionally varying lidar ratios in its aerosol-retrieval algorithm (Haarig et al., 2025), which will improve the accuracy of the Level-2 dust extinction coefficient, an essential input for dust CCNC retrieval. However, our results also highlight the importance of accounting for regional variations in the microphysical properties of dust (and other aerosol types) when updating OMCAM or developing other future algorithms that are used for global CCNC retrieval from spaceborne lidar measurements. Considering such regional variations in dust microphysics is crucial for the broader applications of spaceborne lidar-derived height-resolved CCNC datasets in ACI studies.

# 5. Summary and conclusions

Obtaining the global height-resolved distribution of CCNC and INPC from lidar observations marks a promising pathway for ACI studies. However, the POLIPHON method requires aerosol-type-specific and regional varying conversion factors for transforming optical parameters from lidar measurements into cloud-relevant aerosol concentrations.

Here, we extend our earlier work to obtain an extended dataset of 532-nm dust-related conversion factors at 198 AERONET sites. This includes mass- and INP-relevant conversion factors at 137 sites and CCN-relevant conversion factors at 123 sites.

The geographical distribution of these sites ensures that major deserts and routes of dust transport are now represented by corresponding conversion factors. We find regional variations in dust-related conversion factors that suggest changes in dust microphysical properties along the transport pathways of dust plumes. For instance, differences in the gravitational settling of fine and coarse dust modulate the shape of the PSD during the transport. Moreover, the varying levels of mixing with other aerosols might contribute to regional variations of the conversion factors since our relaxed criterion for identifying dust presence may lead to the inclusion of non-dust particles. In general,  $c_{\rm v,d}$  tends to decrease with greater distance from dust sources. In contrast,  $c_{250,\rm d}$ ,  $c_{\rm s,d}$ , and  $c_{\rm s,100,\rm d}$  are found to be larger downstream of desert regions. The CCN-relevant conversion factors  $c_{100,\rm d}$  and  $\chi_{\rm d}$  show site-to-site variations without a clear regional pattern, because they are more sensitive to the contribution of local fine-mode particles. Overall, our findings highlight the importance of considering geographic variations in dust-related conversion factors for inferring dust-related particle concentrations from lidar observations.

Note that compiling a gridded dust conversion factor dataset is challenging, although such a data set would be highly useful for future studies of global ACI. This arises from the limited number of available sites relative to global coverage, as well as their inhomogeneous geographical distribution. We recommend using values from the nearest available site when applying the current conversion factor dataset.

To test the performance of the derived conversion factors, we conduct a comparison of CALIOP-based dust-related CCNC profiles by applying the POLIPHON and OMCAM methods to data collected at 12 AERONET sites. We generally find agreement within an order of magnitude, which is acceptable given the respective retrieval uncertainties (Choudhury and Tesche, 2023a; Ansmann et al., 2019a). It is most likely that site-to-site variations in dust microphysical properties contribute to these differences. OMCAM employs a single fixed dust PSD from the CALIPSO aerosol model, while POLIPHON uses climatology-based conversion factors that account for regional variations in dust PSD. The most notable difference is that the PSDs of the identified dust data in this study show much higher coarse-to-fine dust particle number ratios compared to the fixed dust PSD used by CALIPSO. This difference contributes to a much higher number concentration of fine particles for OMCAM to reconstruct a similar particle extinction coefficient, and finally leads to higher dust CCNC values compared with POLIPHON. As a consequence, discrepancies in CCNC profiles between the two methods partly reflect the inadequate representativeness of the CALIPSO-model-defined dust PSD at different locations. It is a trade-off for the current version of OMCAM to use the globally fixed, aerosol-type-specific PSDs to retrieve a reasonably accurate CCNC dataset, given the limitations of the current Version 4 CALIOP retrievals. Nevertheless, additional in situ measurements will be essential in the future to validate the capability of both POLIPHON and OMCAM in retrieving global dust CCNC climatology. Therefore, further efforts are needed in incorporating regional-dependent microphysics of dust (and other aerosol types) to improve the OMCAM algorithm, for its broader applicability to ACI studies on a global scale.

We have tested the conversion factors by comparing the derived CCNC profiles with CCNC profiles generated by OMCAM retrievals. In the future, it is also necessary to validate the conversion factor dataset by comparing the retrieved CCNC and INPC (or INP-relevant parameters such as  $n_{250,d}$  and  $s_d$ ) profiles with other independent, co-located, and simultaneous data, from either model outputs (Chatziparaschos et al., 2024; Herbert et al., 2025), in situ measurements (Haarig et al., 2019; Marinou et al., 2019; Kezoudi et al., 2021; Lenhardt et al., 2023), or airborne lidar measurements (Müller et al., 2014). Furthermore, the newly launched EarthCARE ATLID (Atmospheric LIDar) spaceborne lidar also requires conversion factors at 355 nm (Wehr et al., 2023), which can also be calculated with our method. Given the increasing use of ceilometers, extending the conversion factor dataset to a wavelength of 910 nm is also of interest. In addition to dust, conversion factors for other aerosol types (e.g., smoke, volcanic aerosol, sea spray aerosol, anthropogenic aerosol, and so on), as well as their regional-variation features should also be estimated to further extend the applicability of the POLIPHON method in estimating height-resolved CCNC, which is a key parameter to improve our understanding of ACI (Tan et al., 2014; Ansmann et al., 2021; Córdoba et al., 2021).

## Data availability

AERONET data used in this work can be accessed at https://aeronet.gsfc.nasa.gov/ (AERONET, 2023a, 2023b); CALIPSO aerosol profile product can be downloaded at https://subset.larc.nasa.gov/ (CALIPSO, 2025); OMCAM CCNC dataset can be accessed via the link: https://doi.org/10.1594/PANGAEA.956215 (Choudhury and Tesche, 2023b).

#### **Author contributions**

Yun He conceived the research, analyzed the data, acquired the research funding, and wrote the manuscript. Goutam Choudhury analyzed the data, participated in scientific discussions, and proofread the manuscript. Matthias Tesche, Albert Ansmann, and Detlef Müller reviewed the manuscript and participated in scientific discussions. Fan Yi acquired the research funding and led the study. Zhenping Yin participated in scientific discussions and data analysis.

## **Competing interests**

The authors declare that they have no conflict of interest.

## Financial support

This work was supported in part by the National Key Research and Development Program of China under Grant 2023YFC3007802. This research has also been supported by the National Natural Science Foundation of China (grant nos. 42005101, 41927804, and 42205130), the Natural Science Foundation of Hubei Province (grant no. 2023AFB617), the Chinese Scholarship Council (CSC) (grant no. 202206275006), and the Meridian Space Weather Monitoring Project (China). Goutam Choudhury was supported by the German Research Foundation (Deutsche Forschungsgemeinschaft, DFG; grant no. 524386224). Matthias Tesche has been supported by the Federal State of Saxony and the European Social Fund (ESF, grant no. 100649813).

#### Acknowledgements

The authors thank all PIs of the AERONET sites used in this study for maintaining their instruments and providing their data to the community, and the Atmospheric Science Data Central at the NASA (National Aeronautics and Space Administration) Langley Research Center for providing the CALIPSO data.

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
