# Peer review of "Extended POLIPHON dust conversion factor dataset for lidarderived cloud condensation nuclei and ice-nucleating particle concentration profiles"

_EGUsphere, 2025_

## Referee Comment (RC1)

This work tends to provide a regionally adaptable framework for converting observations of space-borne lidars into cloud-relevant aerosol properties, such as CCN and INP. The use of conversion factors, to calculate the bulk particle properties form a single extinction coefficient, when the aerosol type is known, is a promising direction. Authors consider dust-related conversion factors at numerous AERONET sites, which cover major dust transport pathways. They demonstrate that dust-related conversion factors decrease with distance from source regions. The study also compares CCNC from POLIPHON and OMCAM revealing discrepancies between these approaches. This study presents an important contribution to global characterization of CCNs and INPs from space. The manuscript is well written and is suitable for publishing in AMT. I have just some technical comments.

Ln 58. "or a 1020-nm particle linear depolarization ratio >53%". Why it is so high?

Table 1. LR is not introduced.

Ln 123 "We calculate the column-integrated dust ratio…" Would be good to explain meaning of dust ratio.

Eq.3. Subscript 1020 is written not everywhere. Should be harmonized.

Ln 125 "PLDR values are set to 0.30 and 0.02, respectively" In the beginning authors mentioned depolarization above 50%, for this calculation they use 30%. Should be explained.

Ln 131. What is $J_d$?

Table 2. Is it possible to estimate uncertainties of presented parameters?

---

## Referee Comment (RC2)

The paper "Extended POLIPHON dust conversion factor dataset for lidar-derived cloud condensation nuclei and ice-nucleating particle concentration profiles" presents and discusses the dust-related CCN- and INP- conversion factors as extracted using AERONET observations at stations established and operated around the globe. These different conversion parameters are of critical importance for the POLIPHON methodology to be applied in order to assess dust climate impact, at least with respect to clouds and ACI. The study falls within the scope of AMT. The authors have done a thorough job, the manuscript is well-written / structured, the presentation clear, the language fluent and the quality of the figures high. Furthermore, the authors give credit to related work and the results support the conclusions. However, in order to help improving the manuscript, I would kindly suggest the authors to take into account the following comments and recommendations.

1. One of the central component of the analysis is AERONET columnar particle linear depolarization ratio at 1020 nm ($\delta_{1020nm}$), according to my understanding, provided according to the model of randomly oriented spheroids. Thus, I would suggest to discuss on the impact of this assumption. Please provide – maybe as a supplement if do not want to include in the manuscript – a sensitivity study on how the CCN and INP conversion factors change with different AERONET $\delta_{1020nm}$ thresholds considered. For instance here 80% is used. Which would be the change in the case of 70% or 90%, or 95% is used?
2. Since a significant number of PollyXT lidars operate AERONET stations, my suggestion would be include and discuss intercomparison and evaluation of the AERONET-based depolarization ratio against the Polly lidar depolarization ratios, even if it is a different wavelength, under events of dust, polluted dust, dusty marine, and non-dust, in order to strengthen the argument of the suitability of the AERONET-based depolarization ratio to extract CCNC and INPC conversion factors. This comparison will greatly support the argument on the value of 3D CCN and INP dust-related studies globally. Similar studies in the framework of the POLPHON family have been performed, however, the present study claims a global dataset of conversion factors, thus a global implementation of Polly observations can be used to support the value of the dataset to address the climatic effect of dust at a global scale.
3. The authors should go into more details on the variability in dust microphysical properties of dust around the globe, for the main objective is to apply the conversion factors eventually in lidar observations through POLIPHON, possible at regions and conditions of dust transport significantly different than the observed at the specific stations of the present study. The authors should discuss the change of the extracted and proposed CCNC and INP conversion factors as a function of aeolian transport and distance, for aging and mixing with non-dust aerosol subtypes alters the properties of dust, thus affects the proposed conversion factors. For instance, though dust is hydrophobic, polluted dust following long-distance transport in the atmosphere may not be, may be hydrophilic, acting better as CCN than INP. Moreover, the authors discuss deposition of larger dust particles during atmospheric transport. However, depolarization ratio is a function of dust PSD. Applying uniform $d_{dust}$ in decoupling dust and then discussing the removal of coarse dust particles during transport raises thoughts on the impact of the decoupling $d_{dust}$ considered in the methodology and on the impact of CCN and INP factors. More important, the assumption of external aerosol

mixtures is crucial in POLIPHON. Discussing changes in microphysical properties possible related to mixing of different aerosol subtypes is crucial however also raises thoughts. Thus, please also include a discussion on the impact on the external mixing assumption of possible mixing of different aerosol subtypes and what is expected in terms of microphysical properties, possible through AERONET observations, since this is a cornerstone also of the study. How do CCN and INP factors affected? Please discuss.

4. Please discuss the impact of the selected dust LR on the extracted CCN and INP conversion factors. For instance, several studies have demonstrated that over the Atlantic Ocean higher than the CALIPSO applied -applied also in the present study-universal 44 sr dust LR are observed. Which would be the impact of a LR higher, i.e. 45 sr on the conversion factors? This is the case of all deserts around the globe. The dataset applying a universal dust LR of 44 sr makes it suitable for universal studies however, when trying to address a scientific question at a regional scale or running an RTM at a specific set of coordinates the CCN and INP conversion factors that have been established, possible with not proper dust LR, will lead to not suitable conversion factors. Please discuss in the manuscript.

5. The outputs -in order to facilitate studies of CALIPSO- should provide dust related CCN and INP conversion factors over regions not covered by AERONET stations, for instance interconnecting the dust plumes over the oceans with the dust sources. The authors mention "... when applying this conversion factor dataset, we recommend selecting values from the nearest available site". In order to facilitate implementation of the proposed conversion factors to satellite observations at least a geographical dependent clustering over the globe plus with information of the variability has to be provided, accounting the boundary areas for discontinuities. The nearest available site may be not the proper selection or several sites in the proximity to be characterized by very different values.

6. At different parts in the manuscript the authors mention "Only data points with aerosol extinctions exceeding 20 Mm-1 are considered ...", "Note that only data points with aerosol extinctions between 20 Mm-1 and 600 Mm-1 are considered ...", "... only results with the regression coefficient $\chi$ ranging from 0.5 to 1.2 are included ..." without providing a robust -or any- explanation on the selected criteria. Please discuss in the manuscript including references on the selection of the boundaries, and how these selections impact the outcomes. For instance, the lower boundary of dust extinction of 20 Mm-1 may not be insignificant in terms of DOD when integrated in a profile, depending on vertical extend of layers. Though this reference here is columnar, still the reason why not applying "larger than zero values" is no discussed or justified properly. Moreover, the 600 Mm-1 significantly impacts the outcomes over deserts where extreme events may be frequently a norm. Please provide a table with all the assumptions and thresholds considered per implementation step of POLIPHON, or an additional column in Table 1.

7. How does the high/low number of cases affect the uncertainties, variability, and confidence of the conversion factors? Please discuss providing additional input where necessary and a figure showing the number of cases per station.

8. The colorbars / colormaps of figures 4 and 6 should be modified. Please include more colors, since they are not clear for possible readers with related deficiencies (such as myself).

---

## Author Comment (AC1)

**Response to RC1**

**General Comments:**

This work tends to provide a regionally adaptable framework for converting observations of space-borne lidars into cloud-relevant aerosol properties, such as CCN and INP. The use of conversion factors, to calculate the bulk particle properties form a single extinction coefficient, when the aerosol type is known, is a promising direction. Authors consider dust-related conversion factors at numerous AERONET sites, which cover major dust transport pathways. They demonstrate that dust-related conversion factors decrease with distance from source regions. The study also compares CCNC from POLIPHON and OMCAM revealing discrepancies between these approaches. This study presents an important contribution to global characterization of CCNs and INPs from space. The manuscript is well written and is suitable for publishing in AMT. I have just some technical comments.

**Response:** We appreciate the reviewer's thoughtful review and constructive comments. All the comments have been addressed in the revised manuscript, and the responses to each comment are given below.
* * *
**Specific comments:**

**Comments:** Ln 58. "or a 1020-nm particle linear depolarization ratio >53%". Why it is so high?
**Response:** Thank you very much for pointing out this mistake. We have modified the related text to '**a column-integrated dust ratio >53% (based on the 1020-nm particle linear depolarization ratio)**'. (please see L58)

**Comments:** Table 1. LR is not introduced.
**Response:** We have added the introduction as below '**dust lidar ratio (LR)**'. (please see L88)

**Comments:** Ln 123 "We calculate the column-integrated dust ratio…" Would be good to explain meaning of dust ratio.
**Response:** We have added the following sentence '**Within the atmospheric column, $R_{d,1020nm}$ reflects the contribution of dust to the total particle backscatter coefficient of an external aerosol mixture (Tesche et al., 2009).**' (please see L128-130)

**Comments:** Eq.3. Subscript 1020 is written not everywhere. Should be harmonized.
**Response:** We have rechecked Eq. (3) and confirm that its current form is correct. $\delta_{1020nm}^{p}$ is the linear particle depolarization ratio derived from sun photometer measurements. The dust $\delta_{d}^{p}$ and non-dust $\delta_{nd}^{p}$ PLDR values are both set to be constants; since they are not obtained from actual measurements at 1020 nm, we prefer not to include '1020nm' in their subscripts, and we appreciate the reviewer's understanding.

**Comments:** Ln 125 "PLDR values are set to 0.30 and 0.02, respectively" In the beginning authors mentioned depolarization above 50%, for this calculation they use 30%. Should be explained.
**Response:** In introduction section, we intend to mean that 53% is a criterion of column-integrated dust ratio for identifying the dust-presence/dominated data point, instead of a linear particle depolarization ratio value. We have modified the related statement in introduction (please see L58). Therefore, here the set of dust and non-dust PLDR is reasonable. Thank you for pointing this out.

**Comments:** Ln 131. What is Jd?

**Response:** We have added the following description **'where $J_d$ is the number of identified dust-containing data points'.** (please see L139-140)

**Comments:** Table 2. Is it possible to estimate uncertainties of presented parameters?
**Response:** We have added the uncertainties into Table 2.

**References:**

Ansmann, A., Mamouri, R.-E., Bühl, J., Seifert, P., Engelmann, R., Hofer, J., Nisantzi, A., Atkinson, J. D., Kanji, Z. A., Sierau, B., Vrekoussis, M., and Sciare, J.: Ice-nucleating particle versus ice crystal number concentration in altocumulus and cirrus layers embedded in Saharan dust: a closure study, Atmos. Chem. Phys., 19, 15087–15115, https://doi.org/10.5194/acp-19-15087-2019, 2019.

---

## Author Comment (AC2)

**Response to RC2**

**General Comments:**

The paper "Extended POLIPHON dust conversion factor dataset for lidar-derived cloud condensation nuclei and ice-nucleating particle concentration profiles" presents and discusses the dust-related CCN- and INP- conversion factors as extracted using AERONET observations at stations established and operated around the globe. These different conversion parameters are of critical importance for the POLIPHON methodology to be applied in order to assess dust climate impact, at least with respect to clouds and ACI. The study falls within the scope of AMT. The authors have done a thorough job, the manuscript is well-written / structured, the presentation clear, the language fluent and the quality of the figures high. Furthermore, the authors give credit to related work and the results support the conclusions. However, in order to help improving the manuscript, I would kindly suggest the authors to take into account the following comments and recommendations.

**Response:** We appreciate the reviewer's thoughtful review and constructive comments. All the comments have been addressed in the revised manuscript, and the responses to each comment are given below.

**Specific comments:**

**Comments:** One of the central components of the analysis is AERONET columnar particle linear depolarization ratio at 1020 nm ($\delta_{1020nm}$), according to my understanding, provided according to the model of randomly oriented spheroids. Thus, I would suggest to discuss on the impact of this assumption. Please provide – maybe as a supplement if do not want to include in the manuscript – a sensitivity study on how the CCN and INP conversion factors change with different AERONET $\delta_{1020nm}$ thresholds considered. For instance, here 80% is used. Which would be the change in the case of 70% or 90%, or 95% is used?

**Response:** Thank you for the insightful discussions. In AERONET retrieval, the aerosol spheroid model combines the particle size distribution and complex refractive index to compute two elements of the Müller scattering matrix, i.e., $F_{22}(1020nm, 180°)$ and $F_{11}(1020nm, 180°)$. These elements are then used to derive the (backscattering) particle linear depolarization ratio (PLDR) at 1020 nm (Shin et al., 2018):

$$\delta^{p}_{1020nm} = \frac{1 - F_{22}(1020nm, 180°)/F_{11}(1020nm, 180°)}{1 + F_{22}(1020nm, 180°)/F_{11}(1020nm, 180°)}$$

AERONET PLDR data serve as a reliable indicator of dust occurrence and have been validated against lidar-derived values (Noh et al., 2017). Shin et al. (2018) further found that PLDR values at 870 and 1020 nm show better consistency with lidar observations for pure dust particles. However, a detailed evaluation of the sun photometer-derived $\delta^{p}_{1020nm}$ itself is not an easy task and is beyond the scope of this study.

Following the reviewer's suggestion, we conducted a sensitivity analysis by adjusting the threshold for the column-integrated dust ratio used to identify dust-containing data points. Varying this criterion (80%, 70%, and 53%) largely affects the number of AERONET sites with conversion factors available. As expected, a higher dust ratio threshold (e.g., 80%) yields the results of conversion factors that are more representative of pure dust conditions, while a lower threshold (e.g., 53%) increases data availability but include more mixture aerosols. As discussed in the manuscript, selecting an optimal threshold involves balancing data availability and proximity to pure dust conditions. Based on this trade-off, we adopted 80% as criterion. In He et al. (2023), We

have showed that the available dust-containing data points will significantly decrease when using the rigorous criterion of 89% for pure dust situations (Shin et al., 2018), especially when calculating the CCN-related conversion factors. For example, we will almost loss all the sites from South America and Southeast Asia. Additional discussion on this selection has been included in the revised manuscript. (please see L132-135)

[Figure]

Figure 2R. POLIPHON conversion factors for retrieving dust-related CCNC and INPC based on the column-integrated dust ratio criterion of 80% (used in this study).

[Figure]

Figure 3R. POLIPHON conversion factors for retrieving dust-related CCNC and INPC based on the column-integrated dust ratio criterion of 70% (as a sensitivity study).

[Figure]

Figure 4R. POLIPHON conversion factors for retrieving dust-related CCNC and INPC based on the column-integrated dust ratio criterion of 53% (dust-dominated mixture in Shin et al. (2018); as a sensitivity study).

**Comments:** Since a significant number of Polly[XT] lidars operate AERONET stations, my suggestion would be include and discuss intercomparison and evaluation of the AERONET-based depolarization ratio against the Polly lidar depolarization ratios, even if it is a different wavelength, under events of dust, polluted dust, dusty marine, and non-dust, in order to strengthen the argument of the suitability of the AERONET-based depolarization ratio to extract CCNC and INPC conversion factors. This comparison will greatly support the argument on the value of 3D CCN and INP dust-related studies globally. Similar studies in the framework of the POLPHON family have been performed, however, the present study claims a global dataset of conversion factors, thus a global implementation of Polly observations can be used to support the value of the dataset to address the climatic effect of dust at a global scale.

**Response:** To the best of our knowledge, several published studies have already conducted such comparison and validation analyses. We reviewed these studies in our previously published paper, i.e., He et al. (2023), which served as a preliminary test of the method used in the current extended and more comprehensive study. Below we list the relevant excerpts from He et al. (2023).

**'AERONET PLDR data are a good indicator of dust occurrence and have been verified to be well correlated with lidar-derived values (Noh et al., 2017). Shin et al. (2018) found that PLDR values at 870 and 1020 nm are more reliable according to the comparison with those from lidar observations for pure dust particles. Therefore, we use PLDR at 1020 nm $\delta^{\mathrm{p}}_{1020}$ (only denoted as PLDR hereafter) to select the dust-occurring data points for the POLIPHON conversion factor calculation (Shin et al., 2019). Note that the overestimation of near-infrared PLDR is reported by comparison with concurrent polarization lidar observations (Toledano et al., 2019; Haarig et al., 2022), possibly due to the assumption of the spheroid particle in AERONET inversion. Nevertheless, $\delta^{\mathrm{p}}_{1020}$ values are only used to qualitatively identify the dust presence with the presupposed threshold values. Its validity will be verified by comparing the derived conversion factors with those from Ansmann et al. (2019) in Sect. 3.1.'**

Therefore, in our opinion it would be better not to repeat this point, especially considering that the current work is repeatedly described in the text as a follow-up work to He et al. (2023). We would be very grateful for the reviewer's understanding in this regard.

**Comments:** The authors should go into more details on the variability in microphysical properties of dust around the globe, for the main objective is to apply the conversion factors eventually in lidar observations through POLIPHON, possible at regions and conditions of dust transport

significantly different than the observed at the specific stations of the present study.

(1) The authors should discuss the change of the extracted and proposed CCNC and INP conversion factors as a function of aeolian transport and distance, for aging and mixing with non-dust aerosol subtypes alters the properties of dust, thus affects the proposed conversion factors. For instance, though dust is hydrophobic, polluted dust following long-distance transport in the atmosphere may not be, may be hydrophilic, acting better as CCN than INP.

(2) Moreover, the authors discuss deposition of larger dust particles during atmospheric transport. However, depolarization ratio is a function of dust PSD. Applying uniform $d_{dust}$ in decoupling dust and then discussing the removal of coarse dust particles during transport raises thoughts on the impact of the decoupling $d_{dust}$ considered in the methodology and on the impact of CCN and INP factors.

(3) More important, the assumption of external aerosol mixtures is crucial in POLIPHON. Discussing changes in microphysical properties possible related to mixing of different aerosol subtypes is crucial however also raises thoughts. Thus, please also include a discussion on the impact on the external mixing assumption of possible mixing of different aerosol subtypes and what is expected in terms of microphysical properties, possible through AERONET observations, since this is a cornerstone also of the study. How do CCN and INP factors affected? Please discuss.

**Response:** Thank you very much for the constructive comments. During dust transport, the microphysical properties of dust can change in complex ways due to both deposition and internal (aging process) and external mixing with non-dust particles. This complexity is especially relevant given the use of an 80% column-integrated dust ratio as the criterion for identifying dust-containing data points. However, with only sun photometer-based atmospheric column measurements, it is difficult to provide explicit answers to all of the reviewer's questions. Here, we attempt to respond based on our current measurements (AERONET data), supplemented by findings from some previously published studies.

In a trans-Atlantic Saharan dust transport event, Liu et al. (2008) compared the optical properties of dust particles along their transport pathways from Africa to the Gulf of Mexico using CALIOP and NASA Langley Research Center HSRL observations. They found that the particle linear depolarization ratio remained essentially constant (~0.32) throughout the transoceanic transport, demonstrating a notable consistency in dust particle non-sphericity; in contrast, the backscatter color ratio and optical depth ratio (between 1064 nm and 532 nm) showed slight decreases. This suggests that during long-range transport, dust particle deposition can take place without significant evidence of particle aging. Similar results were reported by Yu et al. (2021) during the historic 'Godzilla' dust plume event in June 2020, when Saharan dust was transported to the Caribbean Basin and the southern US.

As for Asian dust, during a mega dust event in March 2021, He et al. (2022) observed unaged, non-spherical dust particles (PLDR >0.3) after their transport (>1000 km) to Wuhan (in central China). In contrast, a statistical study (during 2010-2020) conducted at the same site by Jing et al. (2024) reported average dust PLDR values of 0.14 in spring and 0.11 in winter, both indicating a significant degradation in particle non-sphericity. In winter, dust aerosols over Wuhan tend to reside at lower heights and are more likely to mix with local anthropogenic aerosols, accompanying with relatively moist atmospheric conditions compared with spring. These factors probably contribute to the lower observed PLDR values. This suggests that both external and internal (aging process) mixing can influence the optical and microphysical properties of dust particles.

The abovementioned findings from published studies indicate that changes in dust optical and

microphysical properties during long-range transport can vary significantly from region to region and even from case to case. Therefore, it would be rather difficult to conclude a universal pattern. In the original manuscript, we have tried to discuss this issue whenever possible.

(1) In Section 3, we already discussed several evident variation patterns along major dust transport pathways, for example, '$c_{250,d}$ **near desert regions are relatively lower compared to polluted regions downstream of deserts; a gradual increase in** $c_{250,d}$ **is evident when following the meridional transport of dust from North Africa to Northern Europe, corresponding to the typical northward transport pathway of Saharan dust...'** Similar analyses were also presented in He et al. (2023), focusing on trans-Atlantic and trans-Pacific dust transport pathways. Given the 80% column-integrated dust ratio criterion, there remains a 20% contribution from non-dust components in the atmospheric column. Therefore, mixing with other aerosol types is inevitable. However, with the existing observation data, it is rather difficult to determine definitively whether such mixing is internal or external.

(2) We agree with the reviewer's view that the particle depolarization ratio is a function of the dust PSD. In fact, the POLIPHON method by Mamouri and Ansmann (2014) also proposed a two-step approach that divide the aerosols into non-dust particles ($\delta_p \leq 0.05$), fine-mode dust ($\delta_p = 0.16$), and coarse-mode dust ($\delta_p = 0.39$) (see Figure 5 therein), based on the laboratory measurements by Sakai et al. (2010). As shown, coarse-mode dust solely produces a much higher PLDR than the commonly applied threshold value of 0.3. In addition, Hu et al. (2020) observed the pure dust $\delta_p$ values of 0.37 at 532 nm in the Taklimakan Desert. Therefore, using $\delta_p = 0.3$ as a threshold value remains a valid criterion for identifying pure dust particles (whether fine or coarse mode) within the atmospheric column, according to existing lidar observations, even when some degrees of the coarse-mode dust deposition has occurred (Liu et al., 2008; Yu et al., 2021; He et al., 2022).

(3) This study primarily focuses on retrieving dust-related conversion factors in the POLIPHON method, instead of directly estimating the CCN and INP concentrations. As a result, a detailed discussion of the potential impact of the external mixture assumption on the derived CCN and INP concentrations is somewhat beyond the scope of the current work. As discussed above, distinguishing between internal and external mixing with different aerosol subtypes is highly complex and cannot be adequately addressed using only long-term, global-coverage AERONET data. Such mixing characteristics may vary significantly from region to region, and even from case to case at the same location. From the authors' perspective, a more promising approach would involve conducting comprehensive case studies, integrating multiple measurement techniques, such as ground-based remote sensing, in sit measurements, and spaceborne observations, so as to better understand the effects of aerosol mixing on dust optical and microphysical properties during and after long-range transport. We are grateful for the reviewer's constructive suggestions.

**Comments:** Please discuss the impact of the selected dust LR on the extracted CCN and INP conversion factors. For instance, several studies have demonstrated that over the Atlantic Ocean higher than the CALIPSO applied -applied also in the present study- universal 44 sr dust LR are observed. Which would be the impact of a LR higher, i.e. 45 sr on the conversion factors? This is the case of all deserts around the globe. The dataset applying a universal dust LR of 44 sr makes it suitable for universal studies however, when trying to address a scientific question at a regional scale or running an RTM at a specific set of coordinates the CCN and INP conversion factors that

have been established, possible with not proper dust LR, will lead to not suitable conversion factors. Please discuss in the manuscript.

**Response:** We fully agree with the reviewer's opinion that the dust lidar ratio varies from region to region. Dust particles originating from different deserts can exhibit distinct optical and microphysical properties, such as particle size distribution and complex refraction index. Moreover, variations in dust transport pathways may lead to differences in aging, mixing, and removal processes. These factors contribute to the regional variability in dust lidar ratio values, which generally range from 30 sr to 60 sr (Müller et al., 2007; Tesche et al., 2011; Mamouri et al., 2013; Hofer et al., 2017; Hu et al., 2020; Peng et al., 2021; Floutsi et al., 2023). We have added some relevant sentences in Section 2 to mention this point. (please see L88-92)

In addition, the current Version 4 CALIOP retrieval algorithm uses globally constant, aerosol-type-specific lidar ratios to derive extinction coefficient profiles. For aerosol subtypes such as dust, polluted dust and dusty marine, this approach may be limited by regional variations in dust lidar ratios. Benefitting from in situ and remote sensing measurements collected over the past 15 years, the upcoming Version 5 CALIOP data product is expected to incorporate regionally varying lidar ratios into its aerosol retrieval algorithm (Haarig et al., 2025). This improvement will enhance the accuracy of the Level-2 dust extinction coefficient, a key input for retrieving dust INP and CCN concentrations. We have already discussed this in the last paragraph of Section 4 (please see L344-358). Here we prefer to simply remind readers that this critical issue should be carefully considered in future CALIOP-based retrievals of global dust CCNC and INPC. Since the new version of CALIOP product has not yet been released, it is currently difficult to comprehensively assess the impact of dust lidar ratio selection on the conversion factors at the global scale.

**Comments:** The outputs in order to facilitate studies of CALIPSO- should provide dust related CCN and INP conversion factors over regions not covered by AERONET stations, for instance interconnecting the dust plumes over the oceans with the dust sources. The authors mention "… when applying this conversion factor dataset, we recommend selecting values from the nearest available site". In order to facilitate implementation of the proposed conversion factors to satellite observations at least a geographical dependent clustering over the globe plus with information of the variability has to be provided, accounting the boundary areas for discontinuities. The nearest available site may be not the proper selection or several sites in the proximity to be characterized by very different values.

**Response:** We acknowledge once again that, based on our attempts, it is currently difficult to generate an ideal gridded conversion factor dataset using fewer than 140 AERONET stations with any spatial interpolation methods. A key challenge lies in the highly uneven geographic distribution of AERONET stations. As shown in Figure 2, stations are densely clustered in Europe and North America, which can lead to oversampling when creating a grided dataset. In contrast, station coverage is sparse across other continents and over the oceans, providing inadequate information. Therefore, in the current manuscript, we suggest applying the conversion factors from the nearest available station as a practical solution. In the Figure below, we show a preliminary gridded $c_{s,d}$ derived with Kriging interpolation method as an example. It is clearly seen that sparse geographical coverage of AERNOET sites leads to the failure of acquiring the valid results over the vast oceanic areas.

[Figure]

Figure 4R. Global distribution of gridded $c_{s,d}$ derived with Kriging interpolation method.

Note that the title of this manuscript is '***Extended POLIPHON dust conversion factor dataset for lidar-derived cloud condensation nuclei and ice-nucleating particle concentration profiles***', which does not claim the immediate global applications with satellite data. This reflects the primary objective of the current work is focusing on extending the availability of POLIPHON dust conversion factors to all the possible locations, enabling boarder applications around the world. Therefore, a comprehensive dataset or geographically dependent clustering approach for retrieving global 3-D distributions of dust INPC and CCNC based on spaceborne observations (as proposed/prospected in the current manuscript) would be more appropriately addressed in a dedicated follow-up work. In that future work, reviewer's suggestion of implementing '***geographical dependent clustering over the globe plus with information of the variability has to be provided, accounting the boundary areas for discontinuities***' could be further considered. Accomplishing this would first require establishing clear links between different dust source regions and associated downstream areas, which remains a substantial and complex task. We would be very grateful for the reviewer's understanding in this regard.

**Comments:** At different parts in the manuscript the authors mention "Only data points with aerosol extinctions exceeding 20 Mm$^{-1}$ are considered ...", "Note that only data points with aerosol extinctions between 20 Mm$^{-1}$ and 600 Mm$^{-1}$ are considered ...", "... only results with the regression coefficient $\chi$ ranging from 0.5 to 1.2 are included ..." without providing a robust -or any- explanation on the selected criteria. Please discuss in the manuscript including references on the selection of the boundaries, and how these selections impact the outcomes. For instance, the lower boundary of dust extinction of 20 Mm$^{-1}$ may not be insignificant in terms of DOD when integrated in a profile, depending on vertical extend of layers. Though this reference here is columnar, still the reason why not applying "larger than zero values" is no discussed or justified properly. Moreover, the 600 Mm$^{-1}$ significantly impacts the outcomes over deserts where extreme events may be frequently a norm. Please provide a table with all the assumptions and thresholds considered per implementation step of POLIPHON, or an additional column in Table 1.

**Response:** Thank you very much for pointing this out. Regarding the upper limit of aerosol extinction at 600 Mm$^{-1}$, Ansmann et al. (2019) have already provided a thorough discussion (see Section 3.2 and Figures 4 and 5 therein). They observed that the correlation strength significantly decreases with increasing AOD and becomes indistinct for measurements with AOD values of 1-3

(i.e., aerosol extinction from 1000 Mm$^{-1}$ to 3000 Mm$^{-1}$), with the Dushanbe dataset serving as a representative example. They speculated that the weak relationship for AOD>0.6 (i.e., aerosol extinction >600 Mm$^{-1}$) may be due to the following reasons: **(1)** At very high AOD levels, the coarse-mode dust fraction may dominate the measured optical properties and respective inversion results, making it difficult to reliably retrieval of the particle fraction in the radius range of 100-200 nm. **(2)** More inversion computations are based on AERONET observations taken during early morning and evening hours, when the effective impact of aerosols is strongest (so that the effective dust AOD can be more than twice the column-integrated value stored in the AERONET database). Under such low-visibility conditions, the short-wavelength AERONET channels (340 and 380 nm) may have problems correctly measuring AOD, leading to significant uncertainties in the retrieved $n_{100,d}$. For these reasons, Ansmann et al. (2019) limited the determination of the conversion factor $c_{100,d}$ and $\chi_d$ using regression analysis to AOD values of 0.1-0.6 (i.e., aerosol extinction from 100 Mm$^{-1}$ to 600 Mm$^{-1}$). We have added the related explanations regarding the selection of this lower limit of aerosol extinction in Section 3.

Ansmann et al. (2019) applied a threshold of AOD>0.1 (i.e., aerosol extinction >100 Mm$^{-1}$) when calculating INP- and CCN-related dust conversion factors, in order to exclude cases with very clean atmospheric conditions that may introduce large uncertainties in the computations. In the current study, we attempted to loosen this lower limit to an AOD of 0.02 (i.e., aerosol extinction >20 Mm$^{-1}$) to increase the number of available data points for conversion factor retrieval. Based on the comparisons with Ansmann et al. (2019), we confirm that relaxing this threshold does not significantly affect the results. We have added the related explanations regarding the selection of this lower limit of aerosol extinction in Section 3 (please see L232-237 and L273-275). For clarity, we have added the criteria of aerosol extinction used in calculating the conversion factors to the updated Figure 1.

The values of $\chi_d$ often varies significantly from region to region. According to the experience of Ansmann et al. (2019), $\chi_d$ values typically center around 0.8. However, to remain cautious in our interpretation, we have decided to remove this constraint.

**Comments:** How does the high/low number of cases affect the uncertainties, variability, and confidence of the conversion factors? Please discuss providing additional input where necessary and a figure showing the number of cases per station.

**Response:** From a statistical perspective, we provide the standard deviations of conversion factors to reflect their variability and confidence levels, which already account for both the effects of the number of available data points and the degree of dispersion at each site. All relevant information has been included in the uploaded dataset, which is accessible via the following link: https://doi.org/10.5281/zenodo.15281078 (He, 2025). Regarding uncertainties, we estimate that the INP-related and CCN-related conversion factors carry uncertainties of approximately 20-30% and 50-200%, respectively, as thoroughly analyzed in Ansmann et al. (2019) (see Table 1 therein). We have also added the uncertainties into the updated Table 1.

**Comments:** The colorbars / colormaps of figures 4 and 6 should be modified. Please include more colors, since they are not clear for possible readers with related deficiencies (such as myself).

**Response:** In response to the reviewer's suggestion, we have updated the color bars in Figures 4 and 6 to a 'rainbow'-style color scheme to improve color distinction and enhance readability.

**References:**

Ansmann, A., Mamouri, R.-E., Hofer, J., Baars, H., Althausen, D., and Abdullaev, S. F.: Dust mass, cloud condensation nuclei, and ice-nucleating particle profiling with polarization lidar: updated POLIPHON conversion factors from global AERONET analysis, Atmos. Meas. Tech., 12, 4849–4865, https://doi.org/10.5194/amt-12-4849-2019, 2019.

Floutsi, A. A., Baars, H., Engelmann, R., Althausen, D., Ansmann, A., Bohlmann, S., Heese, B., Hofer, J., Kanitz, T., Haarig, M., Ohneiser, K., Radenz, M., Seifert, P., Skupin, A., Yin, Z., Abdullaev, S. F., Komppula, M., Filioglou, M., Giannakaki, E., Stachlewska, I. S., Janicka, L., Bortoli, D., Marinou, E., Amiridis, V., Gialitaki, A., Mamouri, R.-E., Barja, B., and Wandinger, U.: DeLiAn – a growing collection of depolarization ratio, lidar ratio and Ångström exponent for different aerosol types and mixtures from ground-based lidar observations, Atmos. Meas. Tech., 16, 2353–2379, https://doi.org/10.5194/amt-16-2353-2023, 2023.

Haarig, M., Ansmann, A., Engelmann, R., Baars, H., Toledano, C., Torres, B., Althausen, D., Radenz, M., and Wandinger, U.: First triple-wavelength lidar observations of depolarization and extinction-to-backscatter ratios of Saharan dust, Atmos. Chem. Phys., 22, 355–369, https://doi.org/10.5194/acp-22-355-2022, 2022.

Haarig, M., Engelmann, R., Baars, H., Gast, B., Althausen, D., and Ansmann, A.: Discussion of the spectral slope of the lidar ratio between 355 and 1064 nm from multiwavelength Raman lidar observations, Atmos. Chem. Phys., 25, 7741–7763, https://doi.org/10.5194/acp-25-7741-2025, 2025.

He, Y., Yi, F., Yin, Z., Liu, F., Yi, Y., and Zhou, J.: Mega Asian dust event over China on 27–31 March 2021 observed with space-borne instruments and ground-based polarization lidar, Atmos. Environ., 285, 119238, https://doi.org/10.1016/j.atmosenv.2022.119238, 2022.

He, Y., Yin, Z., Ansmann, A., Liu, F., Wang, L., Jing, D., and Shen, H.: POLIPHON conversion factors for retrieving dust-related cloud condensation nuclei and ice-nucleating particle concentration profiles at oceanic sites, Atmos. Meas. Tech., 16, 1951–1970, https://doi.org/10.5194/amt-16-1951-2023, 2023.

Hofer, J., Althausen, D., Abdullaev, S. F., Makhmudov, A. N., Nazarov, B. I., Schettler, G., Engelmann, R., Baars, H., Fomba, K. W., Müller, K., Heinold, B., Kandler, K., and Ansmann, A.: Long-term profiling of mineral dust and pollution aerosol with multiwavelength polarization Raman lidar at the Central Asian site of Dushanbe, Tajikistan: case studies, Atmos. Chem. Phys., 17, 14559–14577, https://doi.org/10.5194/acp-17-14559-2017, 2017.

Hu, Q., Wang, H., Goloub, P., Li, Z., Veselovskii, I., Podvin, T., Li, K., and Korenskiy, M.: The characterization of Taklamakan dust properties using a multiwavelength Raman polarization lidar in Kashi, China, Atmos. Chem. Phys., 20, 13817–13834, https://doi.org/10.5194/acp-20-13817-2020, 2020.

Liu, Z., Omar, A., Vaughan, M., Hair, J., Kittaka, C., Hu, Y., Powell, K., Trepte, C., Winker, D., Hostetler, C., Ferrare, R., and Pierce, R.: CALIPSO lidar observations of the optical properties of Saharan dust: A case study of long-range transport, J. Geophys. Res., 113, D07207, https://doi.org/10.1029/2007JD008878, 2008.

Jing, D., He, Y., Yin, Z., Liu, F., and Yi, F.: Long-term characteristics of dust aerosols over central China from 2010 to 2020 observed with polarization lidar, Atmos. Res., 297, 107129, https://doi.org/10.1016/j.atmosres.2023.107129, 2024.

Mamouri, R. E., Ansmann, A., Nisantzi, A., Kokkalis, P., Schwarz, A., and Hadjimitsis, D.: Low Arabian dust extinction-to-backscatter ratio, Geophys. Res. Lett., 40, 4762–4766, https://doi.org/10.1002/grl.50898, 2013.

Mamouri, R. E. and Ansmann, A.: Fine and coarse dust separation with polarization lidar, Atmos.

Meas. Tech., 7, 3717–3735, https://doi.org/10.5194/amt-7-3717-2014, 2014.

Müller, D., Ansmann, A., Mattis, I., Tesche, M., Wandinger, U., Althausen, D., and Pisani, G.: Aerosol-type-dependent lidar ratios observed with Raman lidar, J. Geophys. Res., 112, D16202, https://doi.org/10.1029/2006JD008292, 2007.

Noh, Y., Müller, D., Lee, K., Kim, K., Lee, K., Shimizu, A., Sano, I., and Park, C. B.: Depolarization ratios retrieved by AERONET sun–sky radiometer data and comparison to depolarization ratios measured with lidar, Atmos. Chem. Phys., 17, 6271–6290, https://doi.org/10.5194/acp-17-6271-2017, 2017.

Peng, L., Yi, F., Liu, F., Yin, Z. and He, Y.: Optical properties of aerosol and cloud particles measured by a single-line-extracted pure rotational Raman lidar, Opt. Express, 29(14), 21947-21964. https://doi.org/10.1364/OE.427864, 2021.

Sakai, T., Nagai, T., Zaizen, Y., and Mano, Y.: Backscattering linear depolarization ratio measurements of mineral, sea-salt, and ammonium sulfate particles simulated in a laboratory chamber, Appl. Optics, 49, 4441–4449, 2010.

Shin, S.-K., Tesche, M., Kim, K., Kezoudi, M., Tatarov, B., Müller, D., and Noh, Y.: On the spectral depolarisation and lidar ratio of mineral dust provided in the AERONET version 3 inversion product, Atmos. Chem. Phys., 18, 12735–12746, https://doi.org/10.5194/acp-18-12735-2018, 2018.

Tesche, M., Groß, S., Ansmann, A., Müller, D., Althausen, D., Freudenthaler, V., and Esselborn, M.: Profiling of Saharan dust and biomass-burning smoke with multiwavelength polarization Raman lidar at Cape Verde, Tellus B: Chem. Phys. Meteorol., 63, 649–676, https://doi.org/10.1111/j.1600-0889.2011.00548.x, 2011.

Toledano, C., Torres, B., Velasco-Merino, C., Althausen, D., Groß, S., Wiegner, M., Weinzierl, B., Gasteiger, J., Ansmann, A., González, R., Mateos, D., Farrel, D., Müller, T., Haarig, M., and Cachorro, V. E.: Sun photometer retrievals of Saharan dust properties over Barbados during SALTRACE, Atmos. Chem. Phys., 19, 14571–14583, https://doi.org/10.5194/acp-19-14571-2019, 2019.

Yu, H., Tan, Q., Zhou, L., Zhou, Y., Bian, H., Chin, M., Ryder, C. L., Levy, R. C., Pradhan, Y., Shi, Y., Song, Q., Zhang, Z., Colarco, P. R., Kim, D., Remer, L. A., Yuan, T., Mayol-Bracero, O., and Holben, B. N.: Observation and modeling of the historic "Godzilla" African dust intrusion into the Caribbean Basin and the southern US in June 2020, Atmos. Chem. Phys., 21, 12359–12383, https://doi.org/10.5194/acp-21-12359-2021, 2021.